# Mechanical Properties and Microstructure of High-Strength Lightweight Concrete Incorporating Graphene Oxide

**DOI:** 10.3390/nano12050833

**Published:** 2022-03-01

**Authors:** Xiaojiang Hong, Jin Chai Lee, Bo Qian

**Affiliations:** 1Department of Civil Engineering, Faculty of Engineering, Technology and Built Environment, UCSI University, Kuala Lumpur 56000, Malaysia; hxjxcxy@126.com; 2Department of Civil Engineering, Faculty of Civil and Hydraulic Engineering, Xichang University, Xichang 615013, China; qbydf@126.com

**Keywords:** high-strength lightweight concrete, mechanical properties, microstructure, graphene oxide

## Abstract

The increasing demand for high-strength lightweight concrete (HSLWC) with excellent mechanical properties has inspired the development of nanomaterials in fundamentally solving brittleness and cracking. This work investigated the effects of graphene oxide (GO) on the mechanical properties and microstructure of HSLWC, including the workability, density, compressive strength in different curing regimes, splitting tensile strength, flexural strength, modulus of elasticity and scanning electron microscopy (SEM). Six groups of mixtures were mixed with GO aqueous solution at a dosage of 0.00%, 0.02%, 0.04%, 0.05%, 0.06%, and 0.08% by weight of cement, respectively, and dispersed by ultrasound for 30 min. The test results showed that adding a low volume of GO to the specimens could slightly increase the density, rationally reduce the slump, and significantly improve the mechanical properties. The maximum increase in compressive strength, splitting tensile strength, modulus of elasticity and flexural strength of HSLWC with GO at 28 days was by 24%, 17%, 15%, 20%, respectively, as compared with HSLWC without GO. Simultaneously, the SEM results showed that GO could not only fill nano-scale pores, but also regulate the formation and growth of flower-like crystals, which was an important factor for the further improvement of properties. The research results provided a potential new pathway to improve the mechanical properties of HSLWC.

## 1. Introduction

Concrete is considered the most widely used material in construction, and the demand for concrete with more efficient technical specifications is increasing every year [1]. Ordinary concrete is limited in the use of low-bearing capacity areas, super-high building structures and large span structures because of its self-weight, poor ductility and easy shrinkage, and it is imperative to develop and research lightweight aggregate concrete (LWAC) [2]. A 35-story building based on the bearing capacity of the foundation can be raised to 52 stories with LWAC support [3]. Meanwhile, the potential properties of LWAC can reduce energy consumption and pollution in European building development trends [4]. Furthermore, with the advantage of a high strength-to-weight ratio, LWAC can reduce the consumption of steel bars, thus having a positive effect on earthquake sustainability and cost-effectiveness [5].

LWAC refers to concrete with a dry density of less than 1850 kg/m^3^ and performance limitations such as brittleness more prominent than ordinary concrete [6,7,8]. In order to compensate for the defects, the current development direction of LWAC is high strength, especially with excellent mechanical properties. At present, the production of lightweight concrete mainly uses lightweight aggregate (LWA) to replace normal-weight gravel. The traditional LWAs, which include expanded clay, pumice, fly ash, and shale ceramsite, etc., have been used by past researchers [9,10]. Recent research shows that oil-palm-boiler clinker (OPBC), recycled plastic waste and expanded perlite aggregate (EPA) can be used from the perspective of environmental conservation [11,12,13,14]. However, these materials are limited in their mechanical properties. In order to further improve the performance of concrete, High-strength lightweight concrete (HSLWC) was thus produced, which has a dry density of less than 1850 kg/m^3^ and a 28-day compressive strength range of 34–79 MPa [15]. Some researchers have long used fibers as an admixture to produce high-strength lightweight concrete, such as steel fibers, carbon fibers, carbon nanotubes, polymer fibers, etc. [16,17,18]. The use of low-volume steel fiber (≤1%) reduced the workability of concrete, but greatly improved the mechanical properties [19]. The addition of 0.56% polypropylene fiber (by volume of concrete) increased the indirect tensile strength by 90% [20]. In general, the use of fibers may delay the growth of cracks on the micro scale; however, it does not fundamentally delay the nucleation and growth of nano-scale pores and cracks.

Graphene oxide (GO) is a kind of nanomaterial with high strength, good toughness and large specific surface area, which has attracted attention from various fields due to its good physical and mechanical properties [21,22,23]. At present, the study of GO in cement or cement-based materials is still in the exploratory stage, and a relatively unified theory has not been formed. However, some achievements have been made in the influence of the cement matrix hydration process, the mechanical properties mechanism and other aspects. Lv et al. reported that the addition of GO accelerated the hydration process of cement hydration and was the substrate and catalyst for the formation of flower-like crystals [24,25]. In addition, the hydration product could make cement denser, thus improving the mechanical properties of cement-based materials [26]. Lin C et al. found that the -COOH in the GO acted as the “crystal core” to provide a platform for cement hydration and thus accelerated the hydration of cement [27]. Wang M et al. confirmed that GO reacted with Ca(OH)_2_ in cement to produce a 3D network structure [28]. Lin et al. reached a similar conclusion and found that GO provided adsorption sites for water molecules and cement components [29]. Pan Z et al. ascertained that GO increased the surface area of cement slurry and produced more C-S-H, thus preventing the propagation of micro-cracks [30].

In recent years, the preparation process of GO has gradually matured, and a common process is the oxidation of graphite and ultrasonic treatment. Several studies on the addition of low-volume GO to cement and cement-based materials have been developed with precise results in terms of improving mechanical properties. Pan Z et al. also reported that the addition of 0.05% GO by weight of cement which was synthesized from natural graphite using the modified Hummers method could increase the compressive strength of cement slurry by 33% and the flexural strength by 59% [30]. Lv et al. reported that GO with different oxygen content (12.36–29.33%) was prepared by graphite oxidation and ultrasonic treatment. The optimal oxygen content (25.45%) of GO increased the compressive strength, splitting tensile strength and flexural strength of ordinary cement mortar by 60%, 97.2% and 84.5%, respectively [31]. GO prepared by Fakhim Babak et al. had a film thickness of about 20 nm, which confirmed that good adhesion between the GO surface and the surrounding cement mortar resulted in a 48% increase in splitting tensile strength at 1.5% GO content [32]. Li Xueguang et al. reported that the synergistic effect of GO and carbon nanotubes (CNTs) increased the flexural strength of cement composites by 72.7% [33]. It is confirmed that the optimum content of GO that improved the splitting tensile strength in ordinary concrete with a water-cement ratio of 0.5 was 0.03% [34]. When GO was applied to ultra-high strength concrete (UHSC), it was found that GO (0.01% by weight of cement) increased the compressive strength by 7.82% at 28 days [35].

At present, many researchers have focused on the effect of GO on the mechanical properties [36,37], durability and microstructure of cement pastes or mortars, with the results indicating that the addition of a low volume (≤0.1% by weight of cement) of GO could change the crystal morphology of hydration products and greatly improve the properties of cement composites. However, the properties of concrete, especially HSLWC, have rarely been explored. The aims of this paper were to study the fluidity and mechanical properties of high-strength lightweight concrete with different contents of GO. In addition, the morphology characteristics of HSLWC samples were observed by field emission scanning electron microscopy (FE-SEM).

## 2. Materials and Methods

### 2.1. Materials and properties

In this study, ordinary Portland cement type II 42.5R with a specific surface area of 3590 cm^2^/g and specific gravity of 3.14 was used as a binder in all mixes. The coarse and fine aggregates were shale ceramsite (SC) with a nominal maximum size of 19.1 mm and shale pottery sand (SPC) with a fineness modulus of 2.96, respectively. The physical properties of aggregates are shown in Table 1. The GO used in this study had a high-purity single-layer powder with a dark brown color, which was obtained from Suzhou Tanfeng Graphene Technology Co., LTD in China, and its physical and chemical properties are shown in Table 2. A superplasticizer (SP) was based on polycarboxylic ether (PCE) with a water reduction rate of 26.7%.

### 2.2. Mix Proportions

To verify the effectiveness of GO on the mechanical properties of HSLWC, a grade 50 high-strength concrete was designed as the control mix. Typically, the small amounts of GO (about 0.02–0.08% by weight of cement) gave the best gains in the mechanical properties of cement composites [38]. The mix proportions used in this study are shown in Table 3. Therefore, the six groups had the same mix while containing different volumes of GO. Specifically, the GO was added to the G0, G2, G4, G5, G6, and G8 at addition rates of 0.00%, 0.02%, 0.04%, 0.05%, 0.06%, and 0.08%, respectively.

### 2.3. Text Method

The contribution of GO is to achieve uniform dispersion. Due to the poor dispersion effect and agglomeration of GO in cement slurry [39], PCE was used as the active agent to improve the dispersion effect of GO in the alkaline solution [40]. In this study, graphite oxide powder, with 2% superplasticizer aqueous solution, was dispersed by ultrasonication for 30 min to form solutions with six concentrations (GOS solution). The dispersion effect is shown in Figure 1. The contents of GO in Figure 1a–f are 0.00%, 0.02%, 0.04%, 0.05%, 0.06% and 0.08%, respectively. This shows that GO was evenly dispersed without agglomeration. However, when the volume of GO reached 0.08%, GO powder could not be completely dissolved with a few particles. In general, the higher the GOS solution concentration, the darker the color.

All the dried aggregates materials were placed into a pan mixer and mixed for 2 min. Then, the cement was added and mixing continued for another 3 min at moderate speed. After that, 70% of the pre-mixed GOS solution was added into the mixture for a further 3 min. Subsequently, the remaining GOS solution was added and mixed for 2 min before conducting a slump test.

The concrete specimens were cast in the molds to test mechanical properties and microstructure. Slump and density tests were carried out in accordance with the ASTM C143 procedure and the ASTM 138 procedure, respectively, to evaluate the workability of the concrete. In addition, the mechanical tests of every mixture included 100 mm cubes for compressive strength at 1, 3, 7, 28 and 56 days in accordance with the ASTM C39 procedure; cylinders of 100 mm diameter × 200 mm height for splitting tensile strength at 28 days in accordance with the ASTM C469 procedure; prisms of 100 × 100 × 500 mm for flexural strength at 28 days in accordance with the ASTM C78 procedure; and cylinders of 150 mm diameter × 300 mm height for modulus of elasticity at 28 days in accordance with the ASTM C469 procedure. All specimens were compacted in three layers on a vibration table during casting and demolded after 24 h. To obtain valid values, three samples were tested for each testing phase.

In order to investigate the microstructure characteristics, the fracture of the sample was cut into about 10 × 10 × 5 mm after curing in water for 28 days and sprayed with thin gold on the surface of the sample to enhance the electrical conductivity of the sample. Their characteristics regarding crystal morphology were observed with a scanning electron microscope (FE-SEM; thermo scientific Apreo 2C).

### 2.4. Curing Regimes

Concrete should be cured in the curing environment after demolding. The curing environment has a certain influence on concrete performance. Investigating the curing environment of concrete can study the sensitivity of concrete compressive strength in the absence of curing [11,12]. In search of more care and practical curing conditions, the following plans in Table 4 can be considered to study the sensitivity of 28-day compressive strength.

## 3. Results and Discussion

### 3.1. Workability and Density

Table 5 shows the results of the slump tests. It was noted that the high-strength lightweight concrete without GO (G0 mix) has the highest slump value of 95 mm. In addition, with the increase in GO content, the values of slump with GO (G2, G3, G4, G6, and G8) decreased gradually and were lower than that of HSLWC without GO. A higher dosage of GO resulted in lower workability, with a maximum reduction of approximately 40% compared to the mixture without GO. It was also observed that when the added GO was more than 0.04%, the slump of HSLWC varied from 58 to 73 mm. Meanwhile, the light aggregate concrete, with a slump value of 50–75 mm, was similar to that of normal weight concrete, with a slump value of 100–125 mm [3]. Herein, all the mixes showed satisfactory workability.

These results indicate that the incorporation of GO had a negative impact on the workability of HSLWC because of its large specific surface area. A similar conclusion was also reported in previous studies [41]. As can be seen from Figure 2, there was a strong linear correlation between GO content and slump value. This suggested that the slump value at a low volume of GO could be roughly predicted by this figure to facilitate construction use.

Table 5 also shows four types of density in this study, it is noted that with the incorporation of small amounts of GO, the oven dry density of specimens did not increase significantly, at between 1668 and 1756 kg/m^3^. It has been reported that the oven dry density of lightweight concrete was 1440–1840 kg/m^3^ [42]. Therefore, all the mixtures in this study were within this range. Another advantage was that the density change of specimens with GO addition was smaller than that without GO addition, which might be due to GO filling the nanoscale pores and thus reducing water absorption [34]. Owing to these factors, the air dry density of the samples with GO addition was about 9–31 kg/m^3^ lower than the saturated density, while the traditional LWAC was about 100–200 kg/m^3^ [43].

### 3.2. Compressive Strength

#### 3.2.1. Under Full Water Curing

Figure 3 shows the results of compressive strength under full water curing up to 56 days at different GO contents. In general, the compressive strength of specimens with GO was higher than that of those without GO. The 28-day compressive strength of the control mix (G0) under full water curing condition was 53.64 MPa, which was within the range of HSLWC. It was reported that the compressive strength of HSLWC was in the range of 34–79 MPa [15]. Increasing percentages of GO contents from 0.00% to 0.02%, 0.04%, 0.05%, 0.06% and 0.08%, the 28-day compressive strength increased from 53.64 to 66.36 MPa, representing an increase by approximately 4%, 16%, 21% 22% and 24%, respectively. The compressive strength of the specimens increased strongly with GO content until it reached 0.05%, after which no significant enhancement was observed for 0.08%. Such similar behavior was reported by some researchers in cement-based materials [24,34]. This might be caused by the “bridging effect”, “flower crystal formation”, and “increased compactness”, etc.

Figure 4 shows the trend of compressive strength under full water curing with age. It was obvious that GO contributed to the improvement of concrete strength, especially at ages of 3, 7 and 28 days, which might accelerate the cement hydration process. A similar conclusion could be drawn that when GO content reached more than 0.05% (0.05%, 0.06%, 0.08%), the enhancement effect hardly increased. Therefore, among the low volume GO addition rates, 0.05% seems to be the optimal value for compressive strength development in the production of HSLWC.

#### 3.2.2. Under Partial Early Curing

The 28-day compressive strength of HSLWC under different curing regimes are given in Table 6. Under AC and 3W curing regimes, the 28-day compressive strength values of specimens with GO decreased by about 4–8% and 3–6%, respectively, compared with the FW curing regime. However, those without GO declined by 11% and 8%, respectively. There might be two reasons for this: first, keeping in air or water for 2 days was not conducive to complete hydration; second, GO accelerated the hydration reaction and thus impeded the decline [25].

In addition, for specimens without GO (G0 mix), the compressive strength under the 5W curing regime was 97% of that of the FW curing regime, and reached maturity at 7W. Similar reports in the literature suggested that the 7W curing regime was required to achieve maturity strength [44,45]. Moreover, compressive strength of specimen with GO could reach or even slightly surpass the 28 days full water curing when the sample was at 5W. It was reported by Aitcin that the cost of curing was approximately 0.1% to 0.5% of the construction expenditure [46]. Therefore, in order to save construction cost and shorten construction time, the 5W curing regime was allowed to be used for HSLWC mixed with GO in this study.

### 3.3. Splitting Tensile Strength

Figure 5 shows the relationship between GO and the 28-day splitting tensile strength. When the GO content varied from 0.02% to 0.08%, the splitting tensile strength had a strong parabolic fitting relationship, and its trend specifically showed an increase first and then a decrease. Notably, the maximum value was 4.01 Mpa, which occurred when the GO content is 0.05%. Yu-You obtained a similar trend in the study of the ordinary concrete incorporating GO, while the optimal content of GO was 0.03%, which might be due to the water–cement ratio or raw materials [34].

In addition, the 28-day splitting tensile strength of G0, G2, G4, G5, G6 and G8 under full water curing were 3.44, 3.75, 3.82, 4.01, 3.93 and 3.89 MPa, respectively. It was also found that the specimen with GO (G5) had a maximum increase of 17% in 28-day splitting tensile strength, compared with the specimen without GO. The use of light aggregate made the splitting tensile strength of the mixture lower than that of ordinary weight concrete, but it obviously conformed to the requirement of ASTM C330, stating that the minimum splitting tensile strength of a structural light aggregate concrete is 2 MPa [47]. Holm reported that the splitting tensile strength of HSLWC under full water curing was about 6–7% of the compressive strength [48]. In this study, the range was generally 5.9–6.7%. It was previously accepted that there was a necessary relationship between tensile strength and compressive strength, so various empirical types of predictive equations have been proposed.

Figure 6 presents a comparison of predicted splitting tensile strength of HSLWC incorporating GO, which can be calculated using the equations of CEB-FIP [49], BS (blast furnace slag) LWAC [50], Natural Tuff LWAC [51] and OPS LWAC [52]. The measured values enhanced by GO made the results of the three classical prediction equations for light aggregates difficult to accept, but did not exceed the predicted values derived from CEB-PIP. Based on the experimental data of this study, a new equation focusing on the splitting tensile strength of HSLWC enhanced by GO was fitted as follows:f_t_ = 0.45 × f_cu_^0.5^,(1)
where f_t_ is the 28-day splitting tensile strength (MPa), and f_cu_ is the 28-day cube compressive strength (MPa).

### 3.4. Flexural Strength

Table 7 reports the 28-day flexural strength of the six mixtures under full water curing. The specimens containing GO had an increase in flexural strength ranging from 6.1% to 15.0% when GO content varied from 0.02% to 0.08%. But the enhancement effect was not significant as the percentage of GO exceeded 0.05%. Meanwhile, the ratio of flexural strength to compressive strength of concrete at an age of 28 days was between 9.4% and 10.4%. Relevant studies showed that the ratio of HSLWC was approximately 9–11% [53]. Therefore, the ratio in this study was within an acceptable range.

Similar to the splitting tensile strength, the trend also presented a strong parabolic fitting relationship, as shown in Figure 7. It was reported that the flexural strength of LWAC could reach or exceed that of normal weight concrete by adding fibers or nanomaterials [54]. In this study, the mixture with GO had a greater flexural tensile strength than the splitting tensile strength in the range of 54–63%, while the range for normal-weight concrete was 35% [55]. It can be concluded that the effect of low-weight GO content in enhancing the flexural strength is more pronounced than that for the splitting tensile strength.

Figure 8 shows a comparison of predicted values of several equations deriving flexural strength from compressive strength. It was also interesting that the measured value was slightly higher than the predicted value of HSLWC in the literature [43], with a large deviation from the predicted value of other equations. Based on the experimental data of this study, a new equation regarding the flexural strength of HSLWC enhanced by GO was fitted as follows:f_r_ = 0.53 × f_cu_^0.6^,(2)
where f_r_ is the 28-day flexural strength (MPa), and f_cu_ is the 28-day cube compressive strength (MPa).

### 3.5. Modulus of Elasticity

Table 8 shows a comparative analysis of measured values and predicted values derived from different empirical equations for the 28-day modulus of elasticity. It could be seen that the specimens without GO obtained an elastic modulus of 19.88 GPa, which was significantly lower than the normal weight of concrete [42]. According to CEB/FIP, the elastic modulus of light aggregate was in the range of 5–28 GPa, which limited the elastic modulus of LWAC. The specimens with GO achieved a maximum enhancement of 20% compared with those without GO, but still did not compensate for the congenital defect of the low elastic modulus of light aggregates. In addition, it was found from the prediction results that Equations (2) and (3) were in better agreement with the experimental test results, with a prediction error of less than 10%, and there were large errors in the prediction of other equations.

### 3.6. Microstructure

The effect of GO on HSLWC can be further investigated through the analysis of microstructure characteristics as well as mechanical properties. The hydration reaction of cement naturally produces rod-like crystals, needle-like crystals, lamellar crystals and nano-scale pores and other products, which have adverse effects on the mechanical properties of concrete. These crystals were generally assembled from a complex of AFt, AFm, and CH in a disordered state [31]. In contrast, GO can accelerate the hydration reaction of cement and promote the assembly of these regular crystals, such as the formation of denser flower-like crystals. The flower-like crystals tended to be present in pores, cracks, loose hardened paste, and the cracked surface of hardened paste, which might support the evaluation of the improved performance of cement paste [25]. Studies have also shown that the flower-like crystals produced by GO can fill nano-scale pores [22].

Figure 9 shows SEM images of mixtures with different GO contents at 28 days, which are obtained through the investigation of samples randomly selected from mixtures. Large-scale disordered fine needle-like and lamellar crystals could be observed in the specimens without GO, as well as nanoscale gaps between crystals, as shown in Figure 9a. Overall, Figure 9b–f shows that GO can not only fill nano-scale pores, but also regulates the formation and growth of flower-like crystals. The flower-like crystals gradually grew in size and number with increasing GO contents. At GO dosages of 0.02% and 0.04%, it was observed that the specimen was densely filled and formed a few vaguely petal-shaped crystals (Figure 9b,c). At GO dosages of 0.05%, many clusters of crystals clearly appeared perfectly flower-like in shape (Figure 9d). At GO dosages of 0.06% and 0.08%, the clusters of crystals grew more orderly and stronger, and the petals grew thicker. The results suggest that GO can preferentially regulate the formation of flower-like crystals and dense structures. More importantly, at lower dosages, an increase in GO contributed to a better growth environment for flower-like crystals, which might be beneficial for improving the toughness and reducing the cracking of HSLWC.

## 4. Conclusions

Although the fluidity of high-strength lightweight concrete (HSLWC) decreased with the increasing addition of graphene oxide (GO), the test showed satisfactory workability. At a low volume of GO addition, the density of the mixture did not increase significantly, and was still in the range of structural lightweight concrete.With the addition of GO, the maximum increase in compressive strength, splitting tensile strength, flexural strength and elastic modulus of HSLWC at 28 days was 24%, 17%, 15%, 20%, respectively. The results also indicate that 0.05% is the optimum value of GO content for improving the splitting tensile strength of HSLWC, and keeping in water for 4 days after demolding was recommended as an efficient curing regime.Microscopic tests showed that GO not only filled nano-scale pores, but also regulated the formation and growth of flower-like crystals, which might contribute to the further improvement of mechanical properties of HSLWC.The optimal amount of GO in different types of HSLWC and the mechanism of mechanical properties’ improvement still need to be investigated by a large number of experiments, and thus still require further research in the future.

## Figures and Tables

**Figure 1 nanomaterials-12-00833-f001:**
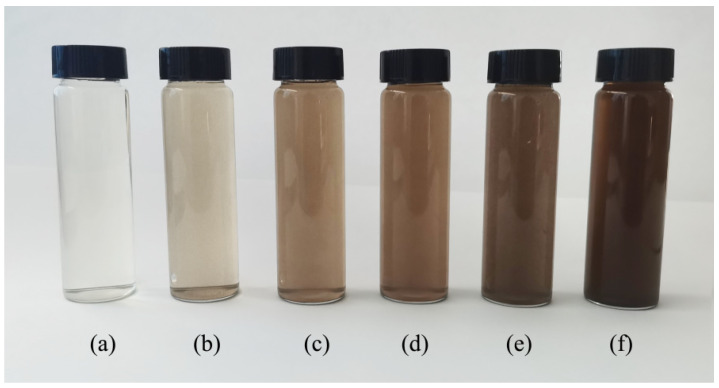
Dispersing effect of different proportions of GO in a 2% superplasticizer aqueous solution: (**a**) 0.00%; (**b**) 0.02%; (**c**) 0.04%; (**d**) 0.05%; (**e**) 0.06%; (**f**) 0.08%.

**Figure 2 nanomaterials-12-00833-f002:**
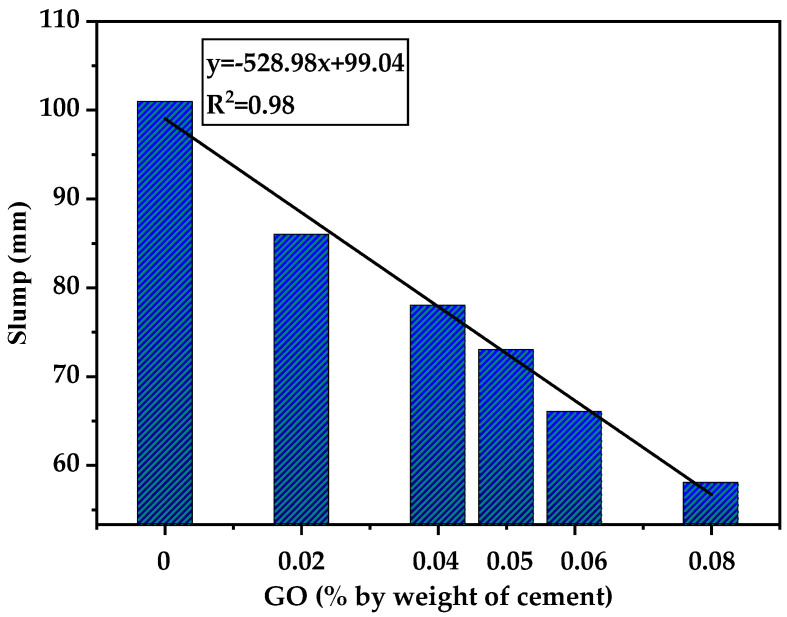
The relationship between slump and density with proportions of GO.

**Figure 3 nanomaterials-12-00833-f003:**
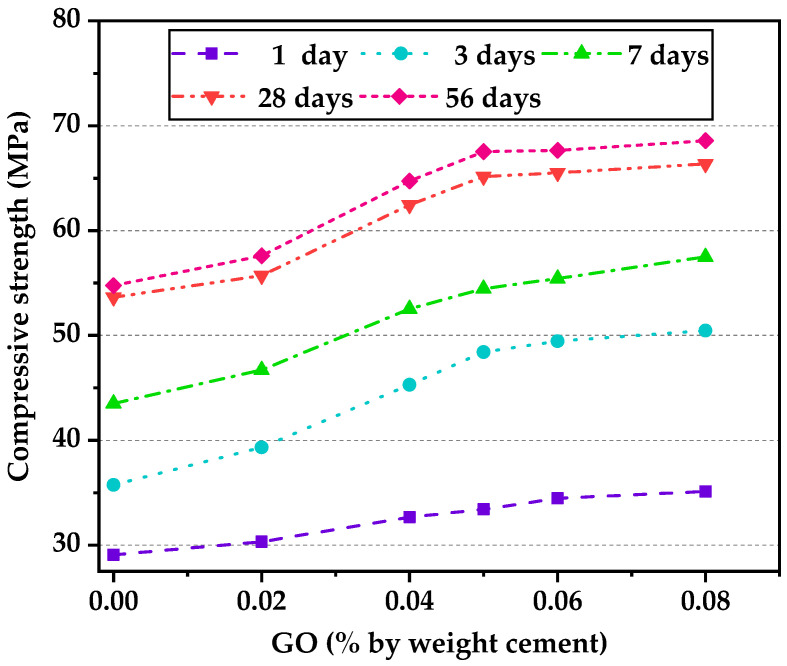
Compressive strength of concrete with varying GO content.

**Figure 4 nanomaterials-12-00833-f004:**
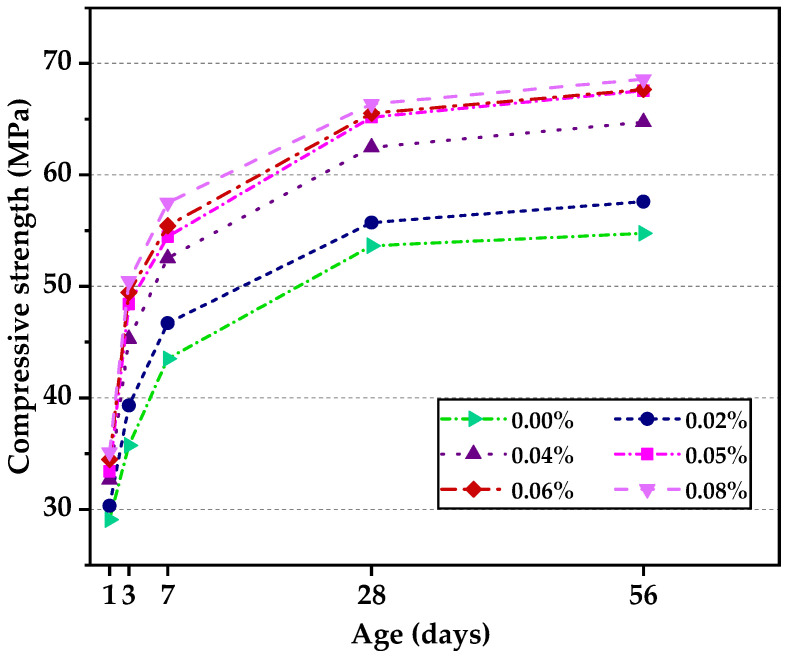
Compressive strength of concrete at different ages.

**Figure 5 nanomaterials-12-00833-f005:**
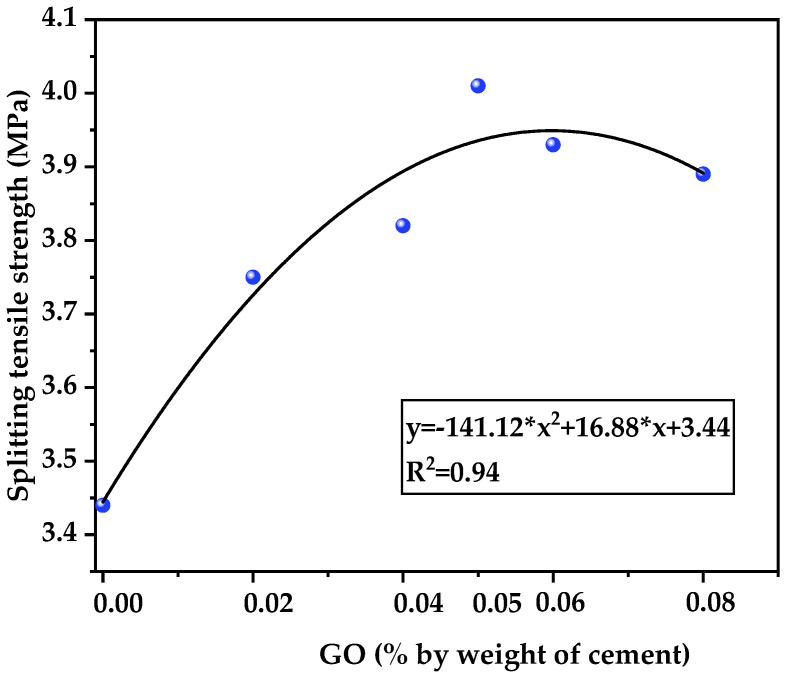
Relationship between GO and the 28-day splitting tensile strength.

**Figure 6 nanomaterials-12-00833-f006:**
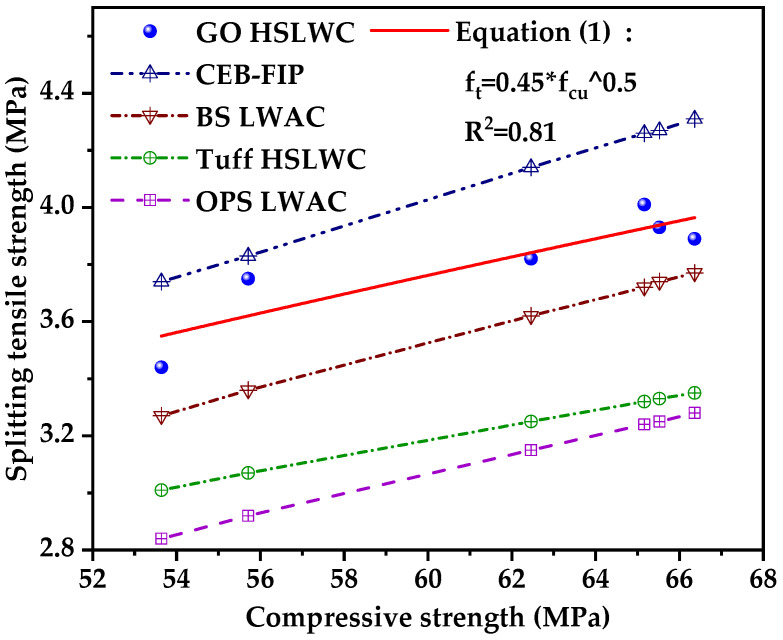
Relationship between the 28-day compressive strength and the 28-day splitting tensile strength of concrete.

**Figure 7 nanomaterials-12-00833-f007:**
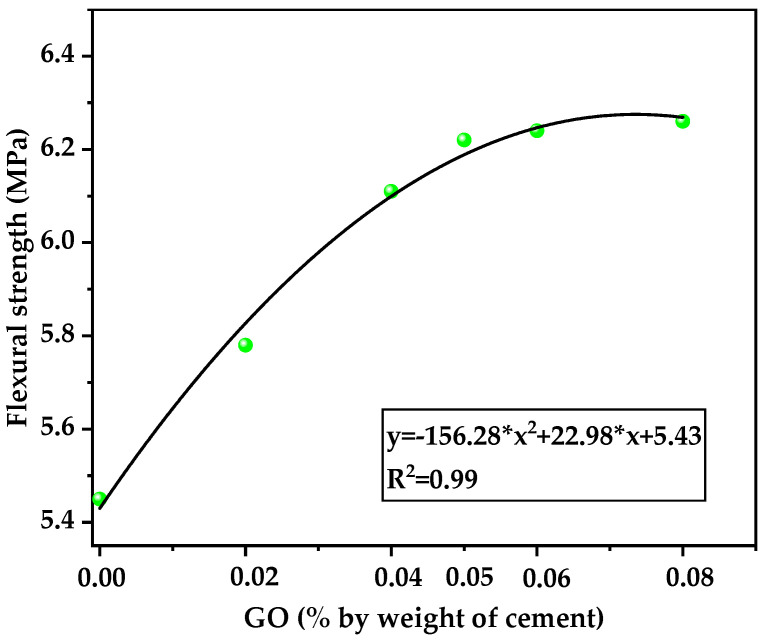
Relationship between GO and the 28-day flexural strength of concrete.

**Figure 8 nanomaterials-12-00833-f008:**
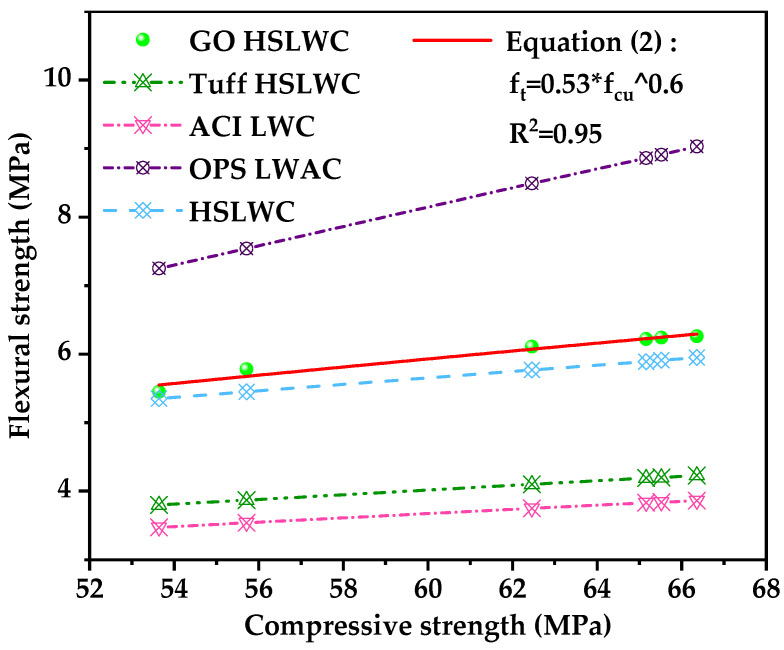
Relationship between compressive strength and flexural strength of concrete.

**Figure 9 nanomaterials-12-00833-f009:**
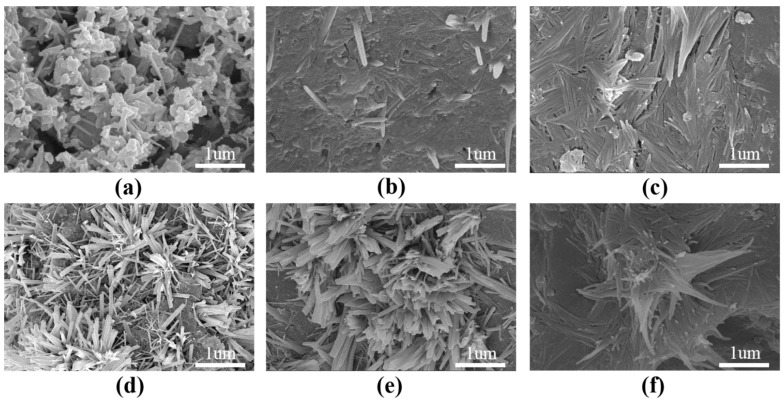
SEM images of the HSLWC with varying GO content at 28 days: (**a**) 0.00%; (**b**) 0.02%; (**c**) 0.04%; (**d**) 0.05%; (**e**) 0.06%; (**f**) 0.08%.

**Table 1 nanomaterials-12-00833-t001:** Physical properties of the aggregates.

Physical Properties	SC	SPC
Fineness	1.93	2.96
Bulk density (kg/m^3^)	835	974
Apparent density (kg/m^3^)	1425	1638
Water absorption (3 h) (%)	2.9	1.23
Water absorption (24 h) (%)	4.6	1.36
Grading Sieve size (mm)	Cumulative % by weight passing	
19.1	100.0	100.0
9.5	86.3	100.0
4.75	9.6	92.0
2.36	0.0	81.0
1.18	0.0	58.0
0.6	0.0	35.0
0.3	0.0	13.6
0.15	0.0	0.0

**Table 2 nanomaterials-12-00833-t002:** Physical and chemical properties of GO.

Physical Parameter						Main Chemical Component (%)		
Specific Surface Area (m^2^/g)	Density (kg/m^3^)	Single Layer Thickness (nm)	Tensile Strength (GPa)	Purity	Number of Layers	C	O	S
232	1780	0.92	0.12	>95 wt.%	5–10	68.44	30.92	0.63

**Table 3 nanomaterials-12-00833-t003:** Mix proportion (per m^3^).

Mix No.	Cement (kg)	Water (kg)	SC (kg)	SPC (kg)	SP (kg)	GO (%) (g)
G0	457	160	520	380	9.1	0 (0)
G2	457	160	520	380	9.1	0.02 (91.4)
G4	457	160	520	380	9.1	0.04 (182.8)
G5	457	160	520	380	9.1	0.05 (228.5)
G6	457	160	520	380	9.1	0.06 (274.2)
G8	457	160	520	380	9.1	0.08 (365.6)

**Table 4 nanomaterials-12-00833-t004:** Curing conditions.

Curing Code	Description of 28-Day Curing Conditions
	The Laboratory and Water Temperature: 23 ± 3 °C
AC	Specimens were placed in air for 27 days after 1 day demolding
3W	Specimens were immersed in water for 2 days after 1 day demolding and then placed in air for 25 days
5W	Specimens were immersed in water for 4 days after 1 day demolding and then placed in air for 23 days
7W	Specimens were immersed in water for 6 days after 1 day demolding and then placed in air for 21 days
FW	Specimens were immersed water for 27 days after 1 day demolding

**Table 5 nanomaterials-12-00833-t005:** Slump and density of concrete mixes.

Mix No.	Slump (mm)	Density (kg/m^3^)			
		Demolded	Air Dry at 28 Days	Saturated at 28 Days	Oven Dry at 28 Days
G0	96	1738	1727	1786	1668
G2	86	1744	1736	1767	1702
G4	78	1749	1745	1764	1732
G5	73	1753	1751	1760	1736
G6	66	1754	1750	1764	1742
G8	58	1768	1763	1777	1756

**Table 6 nanomaterials-12-00833-t006:** The 28-day compressive strength of HSLWC under different curing conditions.

Mix No.	28-Day Compressive Strength under Different Curing Conditions (MPa)				
	FW	AC	3 W	5 W	7 W
G0	53.64	47.77 (89%) ^1^	49.45 (92%)	52.07 (97%)	54.82 (102%)
G2	55.71	51.44 (92%)	52.22 (94%)	56.33 (101%)	57.42 (103%)
G4	62.46	59.50 (95%)	60.22 (96%)	64.41 (103%)	65.87 (105%)
G5	65.16	62.51 (96%)	63.28 (97%)	67.87 (104%)	69.36 (106%)
G6	65.52	62.68 (96%)	63.61 (97%)	68.54 (105%)	69.64 (106%)
G8	66.36	62.84 (95%)	64.44 (97%)	68.67 (103%)	69.78 (105%)

^1^ The data in parentheses are percentages of 28-day compressive strength under FW curing regime.

**Table 7 nanomaterials-12-00833-t007:** Ratio of flexural strength to compressive strength of concrete at an age of 28 days.

Mix No.	f_cu_ (MPa)	f_r_ (MPa)	f_cu_/f_r_ (%)
G0	53.64	5.45	10.2
G2	55.71	5.78	10.4
G4	62.46	6.11	9.8
G5	65.16	6.22	9.5
G6	65.52	6.24	9.5
G8	66.36	6.26	9.4

**Table 8 nanomaterials-12-00833-t008:** The measured 28-day modulus of elasticity and the predicted modulus of elasticity from the equations (MPa).

Mix No.		G0	G2	G4	G5	G6	G8
**Experimental Results**		19.88	22.32	23.53	23.79	23.82	23.88
**Equation No.**	**Ref.**	**G0**	**G2**	**G4**	**G5**	**G6**	**G8**
(3) E = 0.0017w^2^f_cu_^0.33^ *	BS 8110 [56]	18.87 (−5%)	19.31 (−13%)	20.40 (−13%)	20.54 (−14%)	20.70 (−13%)	21.10 (−12%)
(4) E = 0.043w^1.5^f_cy_^0.5^	ACI 318 [57]	20.22 (+2%)	20.76 (−7%)	22.27 (−5%)	22.63 (−5%)	22.79 (−4%)	23.19 (−3%)
(5) E = 0.03w^1.5^f_cy_^0.5^	Hossain et al. [58]	14.10 (−29%)	14.49 (−35%)	15.54 (−34%)	15.79 (−34%)	15.90 (−33%)	16.18 (−32%)
(6) E = 0.04w^1.5^f_cu_^0.5^	CEB/FIP [49]	21.03 (+6%)	21.60 (−3%)	23.16 (−2%)	23.54 (−1%)	23.70 (0%)	24.12 (+1%)
(7) E = (0.062 + 0.0297f_cy_^0.5^)w^1.5^	Slate et al. [59]	18.41 (−5%)	18.83 (−16%)	19.93 (−15%)	20.15 (−15%)	20.28 (−15%)	20.61 (−14%)
(8) E = 2.1684f_cy_^0.535^	Tasnimi [60]	16.20 (−19%)	16.53 (−26%)	17.58 (−25%)	17.98 (−24%)	18.03 (−24%)	18.16 (−24%)
(9) E = 0.0091 (w/2400)^1.5^ f_cu_^2^	Short [61]	13.56 (−32%)	14.78 (−34%)	18.90 (−20%)	20.43 (−14%)	20.77 (−13%)	21.62 (−9%)

* E is the modulus of elasticity (GPa), w is the air dry density (kg/m^3^), f_cy_ is the cylinder compressive strength (MPa) and f_cu_ is the cube compressive strength (MPa). A coefficient of 0.8 proposed by Lo et al. [62] was used to convert the cube to cylinder compressive strength. (+): indicating overestimate, (−): indicating underestimate.

## Data Availability

Data are contained within the article.

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
