# Peer review of "Mechanical Properties and Microstructure of High-Strength Lightweight Concrete Incorporating Graphene Oxide"

_nanomaterials, 2022, doi:10.3390/nano12050833_

Round 1

Reviewer 1 Report

The manuscript "Mechanical properties and microstructure of high strength lightweight concrete incorporating graphene oxide", written by Hong et al., describes interesting approach towards an improvement of properties of HSLWC. The text is well written and majority of data well presented.  I have some comments:

1. HSLWC density on the line 49 seems not to be correct. There are papers declaring values around 1500 kg/m3.
2. Equations used in the manuscript are difficult to read, please use an appropriate equation editor.
3. Quality of the presentation of the table 8 should be improved as the first column is not possible to read.
4. Interesting results are shown in figure 9 (SEM), where considerable changes in the structure are observed. The authors should describe this more in the manuscript. Why is the structure completely different in the figure a, compared to b and c, respectively? Addition of 0.02% of GO into the mixture has only limited effect, as described in the results of some of the tests, but this addition has a great effect on the final structure of the material. Could the authors elaborate on this?

Reviewer 2 Report

The authors present a paper on the use of graphene oxide in the production of high strength lightweight concrete. Overall, the work is interesting and presents a set of relevant results. However, it should be noted that this work does not actually add much to the current state of knowledge.

Some comments are presented below:

The abstract needs to be revised. The abstract must consist of 5 main parts: should mention scope, motivation, methodology, results and expected impact of the research.

In the introduction, when presenting the literature review, it is necessary to mention the influence of the GO properties on the performance of the produced concrete. The chemical composition of the GO can vary significantly by the production process, and thus change the properties of the produced concretes in a significant way.

The presentation of just the physical property of the GO is clearly insufficient. As said, the chemical properties vary significantly depending on the production process and thus influence the produced concretes in a very different way.

In figure 1, the description of the image a) to f) should be in the figure or its caption.

The authors rightly state that “Curing environment has a certain influence on concrete performance”, however, the curind conditions presented in table 4 are not clearly perceived; 2, 4, 6 and 28 days in water? These healing conditions correspond to what? How were they selected? These conditions have nothing to do with, for example, practical conditions of use.

In the description of the experimental campaign, nothing is mentioned regarding the methodology of the tests carried out.

Round 2

Reviewer 1 Report

Authors considerably improved the manuscript and answered my precious comments. I have no further questions.

Reviewer 2 Report

The authors adequately answered all questions and amended the document according to the comments presented. Under these conditions, I believe the paper has minimum conditions to be considered for publication.